# Sigma Receptors: Novel Regulators of Iron/Heme Homeostasis and Ferroptosis

**DOI:** 10.3390/ijms241914672

**Published:** 2023-09-28

**Authors:** Nhi T. Nguyen, Valeria Jaramillo-Martinez, Marilyn Mathew, Varshini V. Suresh, Sathish Sivaprakasam, Yangzom D. Bhutia, Vadivel Ganapathy

**Affiliations:** Department of Cell Biology and Biochemistry, Texas Tech University Health Sciences Center, Lubbock, TX 79430, USA; nhi.t.nguyen@ttuhsc.edu (N.T.N.); valeria.jaramillo-martinez@ttuhsc.edu (V.J.-M.); marilyn.mathew@ttuhsc.edu (M.M.); varshini.suresh@ttuhsc.edu (V.V.S.); sathish.sivaprakasam@ttuhsc.edu (S.S.); yangzom.d.bhutia@ttuhsc.edu (Y.D.B.)

**Keywords:** sigma receptors, progesterone receptor membrane components, labile iron pool, ferroptosis, ferrochelatase, hepcidin, heme chaperone, cytochrome P450, hemochromatosis, cancer

## Abstract

Sigma receptors are non-opiate/non-phencyclidine receptors that bind progesterone and/or heme and also several unrelated xenobiotics/chemicals. They reside in the plasma membrane and in the membranes of the endoplasmic reticulum, mitochondria, and nucleus. Until recently, the biology/pharmacology of these proteins focused primarily on their role in neuronal functions in the brain/retina. However, there have been recent developments in the field with the discovery of unexpected roles for these proteins in iron/heme homeostasis. Sigma receptor 1 (S1R) regulates the oxidative stress-related transcription factor NRF2 and protects against ferroptosis, an iron-induced cell death process. Sigma receptor 2 (S2R), which is structurally unrelated to S1R, complexes with progesterone receptor membrane components PGRMC1 and PGRMC2. S2R, PGRMC1, and PGRMC2, either independently or as protein–protein complexes, elicit a multitude of effects with a profound influence on iron/heme homeostasis. This includes the regulation of the secretion of the iron-regulatory hormone hepcidin, the modulation of the activity of mitochondrial ferrochelatase, which catalyzes iron incorporation into protoporphyrin IX to form heme, chaperoning heme to specific hemoproteins thereby influencing their biological activity and stability, and protection against ferroptosis. Consequently, S1R, S2R, PGRMC1, and PGRMC2 potentiate disease progression in hemochromatosis and cancer. These new discoveries usher this intriguing group of non-traditional progesterone receptors into an unchartered territory in biology and medicine.

## 1. Introduction

Sigma receptors are non-traditional receptors that are not directly coupled to second messengers, like many of the G-protein-coupled receptors, or to gene transcription, like many of the nuclear receptors. They are also not like the growth factor receptors that are associated with tyrosine phosphorylation either. The term “receptor” was assigned to these proteins simply because they bind to a variety of endogenous metabolites and exogenous chemicals with high affinity, often with K_d_ values in the nanomolar-to-micromolar range. The term “sigma” was assigned to the member first identified in this class of proteins because the ligand SKF-10,047 that bound to that protein was a morphine congener whose pharmacological actions could be differentiated from those of the other known morphine (opiate) receptors—mu (μ), kappa (κ), and delta (δ) [1]. Based on the already existing Greek names for the opiate receptors, the new protein that bound SKF-10,047 was called the sigma (σ) receptor simply because of the first letter S in the name of the ligand. Subsequent studies showed, however, that the pharmacological effects of sigma receptor ligands could not be blocked by classical opiate receptor antagonists, such as naloxone [2]. It became clear then that the sigma receptor is not an opiate receptor. Since the features of the binding site in the sigma receptor were found to have some similarities to an already known binding site for phencyclidine, the idea that the sigma receptor could be the same as the phencyclidine binding site was entertained for some time. Even this notion was dispelled subsequently [3]. This led to the definition of the sigma receptor as a non-opiate, non-phencyclidine binding site. Continued research in the area of this newly discovered sigma receptor indicated the existence of two distinct classes of binding sites with overlapping ligand specificities, thus leading to the classification of two different sigma receptors, sigma receptor 1 (S1R) and sigma receptor 2 (S2R) (for reviews, Refs. [4,5,6,7]). Traditionally, the most widely used ligands to differentiate between the two subtypes were (+)-pentazocine for S1R and 1,3-di(2-tolyl)guanidine (DTG) for S2R. As such, (+)-pentazocine binding measured in the presence of DTG is referred to as S1R, and DTG binding measured in the presence of (+)-pentazocine is referred to as S2R. While this definition seems to be fairly correct for S2R, it might not be true for S1R because of the significant overlapping affinity of DTG for both subtypes, which could lead to an underestimation of the S1R binding site. With continued interest in these receptors, several new ligands have now been identified with differential selectivity toward S1R and S2R. In particular, *N*-(4-(6,7-dimethoxy-3,4-dihydroisoquinolin-2(1H)-yl)butyl)-2-(2-fluoroethoxy)-5-iodo-3-methoxybenzamide (RHM-4) has been shown to be far superior to DTG as a selective ligand for S2R in binding studies [8]. Therefore, (+)-pentazocine binding in the presence of RHM-4 rather than in the presence of DTG might be a better strategy for monitoring the S1R binding site. (+)-Pentazocine and RHM-4 are both available in a radiolabeled form to monitor the binding sites selective for S1R and S2R, respectively.

Interestingly, the similarity between S1R and S2R exists only in the sharing of several ligands with overlapping affinities. Successful cloning and the resultant molecular identification of the two receptors led to a surprising revelation—there is no similarity in the primary structure (i.e., amino acid sequence) between the two proteins (Table 1) (reviewed in Refs. [9,10,11]). However, both are integral membrane proteins with one (S1R) or four (S2R) membrane-spanning transmembrane domains. Subsequently, two other proteins were identified, primarily based on ligand-binding features, including the binding of steroids, such as progesterone, that seemed to be related to S1R and S2R, at least at the pharmacological level. These are progesterone receptor membrane component 1 (PGRMC1) and PGRMC2 (reviewed in Refs. [12,13,14,15]). Again, despite the significant overlap in ligands, cloning and the molecular characterization of PGRMC1 and PGRMC2 revealed that the latter two proteins have no structural relationship whatsoever with S1R and S2R (Table 1). However, PGRMC1 and PGRMC2 exhibit a significant similarity between themselves in the amino acid sequence (Table 1). But, S2R has been found to form a complex with PGRMC1, and some of the pharmacological actions assigned to S2R might actually be mediated by this complex. This functional connection and the substantial sharing of the ligands form the basis to group all four proteins under the umbrella term “sigma receptors”. There are several outstanding in-depth reviews on the historical, pharmacological, biological, and structural aspects of these four proteins, authored by experts in this field [9,11,16,17,18,19,20,21].

The goal of this present review is to provide a synopsis of the structural features of these proteins based on the most recent discoveries in the field and then to focus on the interesting roles of these proteins in iron/heme metabolism and homeostasis. The regulatory functions of sigma receptors in the biology of iron and heme are among the latest findings in the field and have not been the major focus of any of the reviews that have been published to date on these proteins.

## 2. Sigma Receptor 1 (S1R)

### 2.1. Amino Acid Sequence and Structure of S1R

S1R was first identified at the molecular level in guinea pig liver [22]. Subsequently, it was cloned from a human placental choriocarcinoma cell line [23], rat brain [24], and mouse [25]. The organization of the human gene coding for S1R has been elucidated [26]. The gene, located in chromosome 9p13, is about 7 kb long and the coding region consists of four exons. The promoter region contains binding sites for the cytokine-responsive transcription factors and also for the xenobiotic-responsive transcription factor (aryl hydrocarbon receptor AhR). The organization of the murine gene has also been elucidated [25]. The human S1R protein consists of 223 amino acids (Figure 1). An analysis of the amino acid sequence using the MINNOU protein transmembrane prediction server [27] predicts the presence of two transmembrane domains in the protein (highlighted in yellow in Figure 1A), five α-helices (identified in red below the sequence in Figure 1A), and nine β-strands (identified in green below the sequence in Figure 1A). At the level of primary structure, S1R has no similarity to S2R, PGRMC1, or PGRMC2, with the identity in the amino acid sequence below 25% (Table 1). Recently, the crystal structures of human S1R [28,29] and *X. laevis* S1R have been determined [30]. Human S1R adopts a homotrimeric configuration (Figure 1B) with each monomer possessing a membrane-spanning transmembrane domain at the N-terminus, followed by a β-barrel body containing the ligand-binding site. The second theoretically predicted transmembrane domain does not traverse the lipid bilayer but lies within the internal leaflet of the lipid bilayer. Each monomer also contains a cupin-like barrel, which houses the ligand-binding site.

S1R binds a wide variety of ligands [31,32], but we focused in this present review on the ability of this receptor to bind heme and progesterone because of the pharmacological relationship of this receptor to S2R and the two progesterone-binding proteins, PGRMC1 and PGRMC2, and also because of the emphasis in this present review on the role of S1R in iron/heme homeostasis. Based on the molecular docking analysis using the AutoDock Vina program, we deduced the theoretical binding energies for progesterone and heme to interact with S1R. The values are −10.1 kcal/mole for progesterone and −9 kcal/mole for heme. This corresponds to K_d_ values of 39 nM for progesterone and 250 nM for heme. This theoretical analysis predicts high-affinity binding of both ligands to S1R. However, to date, only the binding of progesterone to S1R has been demonstrated and studied [33,34,35]. Progesterone binds to S1R with a K_d_ value of 200–400 nM. These values were, however, obtained by indirect means from the dose-dependent competitive inhibition of the binding of S1R ligands by progesterone. When determined directly from the binding of progesterone as the ligand to S1R, the value was 95 nM [36]. It is noteworthy that the experimentally determined K_d_ values for progesterone in different studies are in a similar range to the theoretically derived value. Since S1R is expressed in neural tissues at high levels, it is conceivable that progesterone and other steroids identified as neurosteroids may elicit at least some of their effects in the brain via this receptor [37,38].

The molecular docking analysis indicates a high-affinity binding of heme to S1R, but this feature has not yet been validated experimentally. The theoretically derived K_d_ value for the interaction of S1R with heme (250 nM) strengthens the possibility that S1R could be a heme-binding protein. Progesterone, dehydroepiandrosterone, sphingosine, N,N-dimethylsphingosine, and N-dimethyltryptamine have been proposed as the endogenous ligands for S1R.

### 2.2. Role of S1R in Protection against Neurodegeneration

It has been well established in several studies that S1R plays a protective role in the brain against various forms of neurodegeneration. This includes Alzheimer’s disease, Parkinson’s disease, age-related macular degeneration, diabetic retinopathy, glaucoma, and many other forms of retinal degeneration. Several detailed reviews are available in the literature on this topic [18,39,40,41,42]; readers are referred to these reviews for this topic.

### 2.3. Functional Relationship of S1R to Transcription Factor NRF2

NRF2 is an important transcription factor that regulates gene expression in response to oxidative stress. The ability of this protein to control gene expression is regulated by changes in protein levels, as well as cellular localization. The levels of NRF2 protein can be influenced by changes in expression at the transcriptional level and also by binding to its cytoplasmic partner Keap1 and the resultant ubiquitination and proteasomal degradation. When not bound to Keap1, NRF2 translocates to the nucleus and mediates its effects on the transcription of specific genes by binding to *cis*-elements known as antioxidant-responsive elements present in the promoters of these target genes [43]. Here, we highlight four genes whose transcription is induced by nuclear NRF2; these are glucose-6-phosphate dehydrogenase (G6PD), glutamate-cysteine ligase catalytic subunit (GCLC), glutamate-cysteine ligase modifier subunit (GCLM), and the cystine transporter SLC7A11, as these gene products are directly related to the antioxidant machinery in cells that protects against lipid peroxidation and iron-induced ferroptotic cell death. GCLC and GCLM are involved in the first step in glutathione synthesis, namely the ligation of cysteine to glutamate to form γ-glutamylcysteine, which then is ligated to glycine resulting in glutathione. Cysteine availability in cells is rate-limiting for glutathione synthesis; SLC7A11 provides the cells with this rate-limiting amino acid in the form of cystine, the most prevalent form of cysteine in circulation [44,45,46]. Glutathione is obligatory for the removal of lipid peroxides and hydrogen peroxide via glutathione peroxidases (GPXs). During this step, glutathione (GSH) is converted into oxidized glutathione (GSSG), which needs to be reduced back to GSH to continue the cycle. This reductive step, catalyzed by glutathione reductase, requires NADPH as the electron donor. Hexose monophosphate shunt is the primary metabolic pathway that generates this electron donor, and G6PD catalyzes the first and the rate-limiting step in this pathway. Free iron in ferrous form (Fe^2+^), also known as labile iron, generates hydroxyl radicals (OH^●^) from hydrogen peroxide via the Fenton reaction (Fe^2+^ + H_2_O_2_ → Fe^3+^ + OH^−^ + OH^●^). These hydroxyl radicals oxidize polyunsaturated fatty acids in biological membranes and produce lipid hydroperoxides (ROOH), a process known as lipid peroxidation. The lipid hydroperoxides also undergo the Fenton reaction to generate lipid alkoxyl radical (Fe^2+^ + ROOH → Fe^3+^ + RO^●^ + OH^−^), which perpetuates lipid peroxidation. H_2_O_2_ and ROOH are detoxified with glutathione. As such, under the conditions of excess free iron and deficiency of glutathione, membrane lipids are oxidized to result in a form of cell death called ferroptosis. Since NRF2 maintains cellular levels of glutathione, this transcription factor is directly related to iron biology as a protector of iron-induced ferroptosis.

The activation of S1R with specific ligands, such as (+)-pentazocine, increases the levels of NRF2 protein and NRF2 mRNA in a retinal cone photoreceptor cell line, accompanied by an increase in NRF2-ARE (antioxidant-responsive element) transcriptional activity [47,48]. In a genetic photoreceptor degeneration model in mice, the activation of S1R rescues photoreceptor function, and the effect is obligatorily dependent on the presence of NRF2 [48]. The deletion of S1R results in decreased NRF2 transcriptional activity in retinal Muller cells [49]. In liver cancer cells, oxidative stress induced by inhibitors (erastin, sorafenib) of the cystine transporter SLC7A11 increases S1R protein but without any change in S1R mRNA [50]. These results show a functional crosstalk between S1R and oxidative stress; the activation of S1R prevents oxidative stress by inducing antioxidant response via NRF2 and, at the same time, the induction of oxidative stress increases S1R protein levels. It is important to note that the S1R-dependent NRF2-ARE transcription activity increases the expression of SLC7A11 and GPX4, both proteins being critical for the glutathione-mediated removal of lipid peroxides. Interestingly, the expression of S1R is negatively controlled by NRF2; a decrease in cellular levels of NRF2, or the inhibition of NRF2 with pharmacological agents, is associated with an increase in S1R protein levels [50]. This suggests an effective feedback regulation between S1R and NRF2; the activation of S1R positively controls NRF2 expression and, conversely, NRF2 negatively controls S1R expression.

### 2.4. Protection against Ferroptosis by S1R and Its Relationship to Hemochromatosis and Cancer

The functional interaction between S1R and NRF2-ARE transcriptional activity is directly related to iron homeostasis and ferroptosis. Oxidative stress increases the levels of the labile iron pool and decreases the levels of glutathione in cells with a resultant induction of ferroptosis; this iron-induced cell death process is accelerated by the knockdown of S1R [50]. This shows that S1R protects against ferroptosis, which is supported further by the findings that the knockdown of S1R increases the labile iron pool and the lipid-peroxidation marker MDA (malondialdehyde) [50]. Results similar to the knockdown of S1R are also seen when cells are treated with pharmacological agents that function as antagonists of S1R, such as haloperidol [51].

Excess iron and iron-induced ferroptosis have a connection to several diseases, particularly hemochromatosis and cancer. Hemochromatosis is a genetic disorder of iron overload [52,53], the most prevalent single-gene disease among Caucasians and Hispanics [54]. This disorder is associated with an age-dependent accumulation of iron in multiple systemic organs. Even though hemochromatosis is a genetic disease, clinical symptoms resulting from the excessive accumulation of iron appear only after decades of life. It is surprising that cellular damage does not occur in this disease at a much earlier stage. How do tissues that accumulate excess iron in this disease escape ferroptosis? This biological conundrum is also apparent in cancer. Iron is critical for various cellular functions that are obligatory for cell proliferation, and, accordingly, cancer cells find ways to accumulate iron to support their growth [55,56]. How do cancer cells manage to increase iron levels without being subjected to ferroptosis? It is obvious that hemochromatosis and cancer must be associated with an increase in antioxidant machinery to prevent iron-induced lipid peroxidation and ferroptosis. It is already known that the expression and activity of the cystine transporter SLC7A11 are increased in hemochromatosis and cancer [57,58], which is expected to increase cellular levels of glutathione and provide protection against lipid peroxidation and ferroptosis. These findings highlight the potential role of S1R in these diseases. Several studies have demonstrated a tumor-promoting role for S1R [59,60]. If S1R protects cells from ferroptosis, the tumor-promoting effect of this receptor makes sense. It is important to point out here, however, that the S1R-ferroptosis axis is not likely to be the sole basis for the ability of this receptor to support tumor growth. This receptor is known to regulate a plethora of cellular functions, including mitochondrial function, unfolded protein response, autophagy, and cholesterol metabolism, among others, all of which play a dynamic role in cancer cells. Protection against ferroptosis is yet another important function of S1R that might be critical for the survival of cancer cells, particularly in light of the fact that cancer cells are obligated to accumulate iron to support their rapid proliferation and growth. Given these findings in the field of S1R, it is intriguing to note that there have been no studies reported in the literature on the status of S1R expression and activity in hemochromatosis, the prototypical iron overload disorder.

## 3. Sigma Receptor 2 (S2R)

### Amino Acid Sequence and Structure of S2R

It is important to begin this section with the statement at the onset that sigma receptor 2 (S2R) is not the same as progesterone receptor membrane component 1 (PGRMC1) [reviewed in 9–11,14,15]. This is necessary because of several publications in the literature that claimed PGRMC1 to be S2R [61,62,63]. There is no doubt that a functional relationship exists between the two proteins, but these two proteins are distinct at the molecular level. The actual molecular identity of S2R was not known until 2017, more than 20 years after the cloning of S1R. It was Alon et al. [64] who were successful in cloning S2R and showed that S2R is not PGRMC1 but is instead identical to an already known protein called TMEM97 (transmembrane-protein 97) or MAC30 (meningioma-associated protein 30). S2R consists of 176 amino acids; it belongs to a family of proteins in which the prototypical member is the emopamil-binding protein (EBP). However, unlike EBP, which possesses steroid isomerase activity, S2R does not possess any enzymatic activity. S2R has four transmembrane domains and three small stretches of β-strands (Figure 2A). The POLYVIEW-2D protein structure visualization server [65] was used to predict the transmembrane domains. The AlphaFold model of the amino acid sequence, as per analysis using the Robetta server, yielded a monomer with four transmembrane domains. However, a recent report on the crystal structure of bovine S2R has shown the protein to exist as a homodimer (PDB: 7MFI) [66]. Therefore, the AlphaFold model of human S2R was superimposed onto the structure of bovine S2R to generate the homodimer model for human S2R (Figure 2B). The membrane boundaries were predicted with the OPM (Orientations of Proteins in Membranes) server [67]. S2R is an integral protein of the endoplasmic reticulum, but it translocates to other sites in the cell to form protein–protein complexes in the plasma membrane, as well as in the lysosomal membrane. The gene coding for this protein is 9.5 kb long and is located in chromosome 17q11.2. A review of the molecular, pharmacological, and biological aspects of S2R was recently published by Izzo et al. [68] as the proceedings of an international symposium on this receptor. The biology of S2R is connected to a broad spectrum of cellular functions, including cholesterol transport and metabolism, progesterone signaling, autophagy, and membrane-bound protein trafficking. A notable feature of S2R is that it bears no similarity in amino acid sequence to S1R (Table 1) despite the fact that both proteins are identified as the two subtypes of sigma receptor. As already mentioned earlier in this review, the subtype classification into S1R and S2R was completed solely based on ligand binding long before the molecular identities of the two proteins were established.

A wide variety of pharmacological agents bind to S2R [31,69]. As for the endogenous ligands for this protein, the most likely candidate is the oxysterol known as 20(S)-hydroxycholesterol [70]. As discussed below, one of the well-established biological functions of S2R is its involvement in cholesterol homeostasis. Therefore, it makes sense that one of the metabolites of cholesterol functions as an endogenous ligand for this receptor. In addition, the expression of S2R appears to be under the control of the sterol-dependent transcription factor SREBP-2 (sterol regulatory element binding protein-2) [71].

All known biological functions of S2R seem to be mediated by protein–protein interactions with other proteins (see below). Since S2R is expressed in the brain, and some of its functions are related to the clearance of amyloid-β, there is a growing interest in the potential of this receptor and its ligands in the treatment of Alzheimer’s disease (reviewed in Ref. [21]). For this current review, however, we focused on the ligands heme and progesterone. Surprisingly, we found no published reports in the literature on the interaction of either of these ligands with the cloned S2R. Therefore, we used the molecular docking approach to evaluate theoretically the binding of heme and progesterone with the S2R protein. This analysis yielded a value of −7.1 kcal/mole for the binding energy for heme, which translates to a K_d_ value of 6.2 μM, indicating a low affinity for the interaction. However, it might be appropriate to have this relatively low affinity if S2R functions as a heme chaperone, like PGRMC1/2 (see below). The value for the binding energy for the interaction of progesterone is −7.9 kcal/mole, which corresponds to a K_d_ value of 1.6 μM. This theoretically derived dissociation constant for progesterone binding is significantly higher than the corresponding value for S1R, which has been shown to bind progesterone experimentally. These values predict a lower affinity for progesterone binding to S2R than to S1R. Therefore, it would be of interest to determine if the cloned S2R actually binds progesterone.

## 4. Progesterone Receptor Membrane Components 1 and 2 (PGRMC1 and PGRMC2)

### 4.1. Amino Acid Sequences and Structures of PGRMC1 and PGRMC2

PGRMC1 and PGRMC2 are closely related proteins in the amino acid sequence, with approximately 60% identity (Table 1). But, they do not bear any significant sequence similarity to either S1R or S2R. PGRMC1 contains 195 amino acids and PGRMC2 contains 247 amino acids. Both proteins possess a single membrane-spanning transmembrane domain, highlighted in yellow in Figure 3A. PGRMC1 is an integral membrane protein present in the plasma membrane, mitochondrial membrane, and the membrane of the endoplasmic reticulum. PGRMC2 is also an integral membrane protein and is found in the nuclear membrane and in the membrane of the endoplasmic reticulum. The gene coding for PGRMC1 is located in the X chromosome (Xq24). PGRMC1 is a hemoprotein; the heme in PGRMC1 is penta-coordinated, and Tyr113 serves as the fifth axial ligand for iron in heme (iron in heme is already coordinated to nitrogen; one each in the four pyrroles of protoporphyrin IX). This leaves the sixth coordination surface of heme open, which allows the heme–heme hydrophobic stacking of two heme-containing monomers (Figure 3B) [62]. The resultant homodimer also forms a disulfide link with Cys129, but this covalent linking is not obligatory for dimer formation. The dimerization of heme-bound PGRMC1 has been authenticated with the deduction of its crystal structure [62]. The heme-dimerized PGRMC1 interacts with the EGF receptor [62]. Recent studies by Kabe et al. [72] have identified certain naturally occurring compounds (e.g., glycyrrhizin) that specifically bind to heme-dimerized PGRMC1 and interfere with the interaction of the PGRMC1 dimer with an EGF receptor, with functional consequences in terms of chemoresistance in colon cancer cells. PGRMC2 also binds heme; theoretical modeling, according to the AlphaFold program, suggests a monomeric structure (Figure 3B). In both proteins, the region that is not associated with the membrane contains α-helices and β-strands. The gene coding for PGRMC2 is located in chromosome 4q28.2. The binding of heme, as well as progesterone, to PGRMC1 and PGRMC2, has been established experimentally.

### 4.2. Common Structural Features in PGRMC1 and PGRMC2

Among the four proteins that form the focus of this present review, only PGRMC1 and PGRMC2 are structurally similar. Both bind heme and progesterone. These two proteins are not only similar in amino acid sequence but also share a homologous cytochrome b5-like heme/steroid binding domain [73,74]. There are two other proteins that possess this domain: neudesin and neuferricin. However, unlike PGRMC1 and PGRMC2, which are integral membrane proteins, neudesin and neuferricin are secreted proteins. Because of their ability to bind progesterone, and their feature as integral membrane proteins, PGRMC1 and PGRMC2 are called membrane-associated progesterone receptors to distinguish them from the classical progesterone receptors that function as transcription factors and are not associated with membranes. Even though S1R binds progesterone, may even interact with heme, and is an integral membrane protein, it does not possess the cytochrome b5-like domain. The same is true with S2R. Therefore, S1R and S2R are not members of the membrane-associated progesterone receptor family.

## 5. Protein–Protein Interactions and Functional Complexes among S2R, PGRMC1, and PGRMC2

One of the most widely known biological functions of PGRMC1 is related to its ability to interact with a number of cytochrome P450s and modulate their catalytic activity [13]. The cytochrome P450s that are subject to PGRMC1-mediated regulation are involved in steroid hormone synthesis, cholesterol synthesis, and drug metabolism [13]. The consequence of this protein–protein interaction is not uniform in terms of the enzymatic activity of the involved cytochrome P450s; some are activated and others are inhibited. As mentioned previously, PGRMC1 forms a dimer when bound to heme; there is evidence, at least in the case of specific cytochrome P450s, that it is this dimer of heme-bound PGRMC1 that participates in the interaction with cytochrome P450s [62]. Since the iron in PGRMC1-bound heme is a penta-coordinate resulting from the interaction with nitrogen of each pyrrole ring of the 4-pyrrole-porphyrin and with Y113 of PGRMC1, the heme-iron (i.e., Fe^2+^, not Fe^3+^) is still free to bind carbon monoxide (CO) as the sixth coordinate. This binding of CO to heme in PGRMC1 interferes with dimerization via the blockade of heme–heme stacking. The interaction of PGRMC1 with certain cytochrome P450s has been shown to be inhibited in the presence of CO, suggesting the involvement of the dimer, not the monomer, in this interaction. Similarly, the mutation of Y113 in PGRMC1 disrupts heme binding and, hence, dimer formation, which consequently blocks the interaction of mutated PGRMC1 with cytochrome P450s. Interestingly, the heme-dimerized PGRMC1 may not be the only form by which PGRMC1 influences the activities of cytochrome P450s. There is evidence indicating that PGRMC1 also binds to and modulates the activities of cytochrome P450s in a heme-independent manner [75,76]. These studies have identified 13 different cytochrome P450s that interact with PGRMC1. In comparison to what is known with PGRMC1, relatively much less information is available on the role of PGRMC2 in the regulation of cytochrome P450s. There is, however, evidence for the modulation of the activity of CYP3A4, one of the most robust drug-metabolizing enzymes associated with the endoplasmic reticulum, by PGRMC2 [77,78]. It is likely that PGRMC2 is not involved in the regulation of the cytochrome P450s associated with the synthesis of cholesterol and steroid hormones because these metabolic pathways occur in the mitochondrial membrane and PGRMC2 seems to be absent in this membrane.

Based on the ability of the heme-bound PGRMC1 to interact with CO, this protein has been proposed to function as a CO sensor [79,80]. This newly proposed function of PGRMC1 raises an important issue as to the molecular mechanism by which the transition between Fe^3+^ and Fe^2+^ in PGRMC1-bound heme is accomplished. This is a critical issue that has not yet been addressed. Heme-Fe^2+^ binds CO, whereas heme-Fe^3+^ does not. The binding of molecular oxygen to heme-iron also has a similar requirement. In the cytochrome a3 associated with Complex IV in the electron transport chain, heme-iron oscillates between Fe^3+^ and Fe^2+^ depending on whether or not the electron transfer to O_2_ has been accomplished. When an electron is transferred from heme-Fe^2+^ to heme-bound O_2_, iron in heme transitions from Fe^2+^ to Fe^3+^, which then cannot bind O_2_ anymore. When an electron is delivered from cytochrome c to Complex IV, heme-Fe^3+^ accepts the electron and, thus, becomes heme-Fe^2+^, which then binds O_2_ for the next cycle of electron transfer. In contrast to cytochrome a3, heme in hemoglobin is not involved in electron transfer but still binds O_2_ because of its primary role as the oxygen carrier in circulation. Therefore, heme-iron in hemoglobin has to be present as Fe^2+^ almost all the time. The accidental transfer of electrons to heme-bound O_2_ does occur, hence leading to the conversion of heme-iron to Fe^3+^ and resulting in methemoglobin, which cannot bind oxygen. In order to maintain the function of hemoglobin as an oxygen carrier, there are mechanisms that convert heme-Fe^3+^ in methemoglobin to oxygen-carrying hemoglobin (i.e., heme-Fe^2+^). Two such mechanisms are well known: NADH-dependent cytochrome b5 reductase and NADPH-dependent methemoglobin reductase [81]. It is tempting to speculate that the same or similar mechanisms might operate to mediate the transition of heme-Fe^3+^ to heme-Fe^2+^ in PGRMC1-bound heme in the phenomenon related to the function of PGRMC1 as a CO sensor. It is interesting to note that the PGRMC1 protein possesses a region with a structure similar to that of cytochrome b5, which could be relevant to the issue. Obviously, additional research is needed in this area.

PGRMC1 binds S2R and LDLR to form a complex to regulate the uptake of cholesterol and the clearance of amyloid-β, the latter requiring interaction with ApoE as an additional partner in the complex [82]. There is no evidence for the interaction of PGRMC1 with either S1R or PGRMC2. S2R also interacts with proteins other than PGRMC1 and PGRMC2.

## 6. Biological Functions of S2R, PGRMC1, and PGRMC2

### 6.1. Function of S2R in Cholesterol Homeostasis and Its Potential Connection to Hemochromatosis and Cancer

One of the well-documented and well-studied biological functions of S2R is related to its role in cholesterol homeostasis. This occurs at three levels: cholesterol synthesis, cholesterol uptake in the form of LDL, and cholesterol release from lysosomes following autophagy or LDLR-mediated endocytosis of LDL (reviewed in Refs. [10,11,13,20,21]). The biosynthesis of cholesterol is controlled by SREBP-2, which induces the expression of HMG-CoA reductase, the rate-limiting enzyme in cholesterol synthesis, and the expression of LDLR. The same transcription factor also induces the expression of S2R [71]. S2R forms a complex with LDLR to facilitate cholesterol uptake into cells by promoting the receptor-mediated endocytosis of LDL; this process requires PGRMC1 to form a functional heterotrimeric complex comprising S2R, LDLR, and PGRMC1 [83]. S2R also complexes with the lysosomal cholesterol exporter NPC1 (Niemann–Pick cholesterol transporter) to influence the stability of the NPC1 protein; this results in a negative correlation between S2R and NPC1 protein levels [84]. In other words, S2R decreases the levels of NPC1 and, thus, decreases the export of cholesterol from lysosomes. Since mutations in NPC1 are responsible for the lysosomal storage disease with cholesterol accumulation within lysosomes, it has been suggested that blocking the function of S2R might have a therapeutic benefit in patients with this disease [84].

The regulation of LDLR function by S2R has direct relevance to Alzheimer’s disease. LDLR in brain cells (neurons, astrocytes, microglia) also serves as the receptor for apolipoprotein E (ApoE), which has a role in the cellular uptake of amyloid-β monomers and oligomers [85]. ApoE binds amyloid-β, and the resultant complex is internalized into cells via the cell-surface LDLR. Since the S2R/PGRMC1/LDLR complex formation is necessary for the optimal internalization of LDLR ligands, such as LDL and ApoE, the potentiation of S2R could be therapeutically beneficial for the treatment of Alzheimer’s disease [21]. In fact, S2R has been proposed as a therapeutic target for the treatment of a variety of neurodegenerative diseases, including not only Alzheimer’s disease but also Parkinson’s disease and retinal diseases, such as the dry form of age-related macular degeneration [11,21].

Based on the role of S2R in the regulation of cholesterol homeostasis at multiple levels, S2R is often called a cholesterol-responsive receptor. The function of S2R as a key regulator of cholesterol levels in cells could be relevant to cancer because the need for cholesterol in cancer cells is high to support membrane biogenesis in conjunction with their high rate of proliferation. The connection between S2R and cancer might involve the p53-SREBP-2 axis. The expression of SREBP-2 is under the negative control of the tumor suppressor p53 [86]. Most cancers are associated with decreased levels and/or decreased activity of p53, thus increasing the expression of SREBP-2, which increases the cellular levels of cholesterol. Interestingly, p53 is a heme-binding protein, and the binding of heme subjects p53 to proteasomal degradation [87]. Therefore, any pathological condition that leads to an increase in cellular levels of heme is likely to be associated with decreased levels of p53. An example of this scenario is the iron-overload disease hemochromatosis in which multiple organs are overloaded with iron and heme. With a mouse model of hemochromatosis, it has been shown that the increase in heme levels leads to decreased levels of p53 and the promotion of cancer [88]. Several studies have reported an upregulation of S2R in cancer and suggested the blockade of the S2R function as a potential therapeutic strategy for cancer treatment [89,90]. Since an increased accumulation of iron and heme is a hallmark not only in hemochromatosis but also in cancer, the upregulation of S2R might be a common phenomenon in both cases with the involvement of the heme-p53-SREBP-2-S2R axis as a common underlying mechanism.

S2R also interacts with another integral membrane protein called TSPO (translocator protein), also known as the peripheral benzodiazepine receptor [91]. It would be of interest to note here that, just like PGRMC1, TSPO was also once thought to be S2R. The protein–protein complex formation of S2R with PGRMC1 and TSPO could have contributed to this confusion in the molecular identity of S2R. Very little is known at present about the biological significance of the S2R–TSPO complex. It has been suggested, however, that the interaction of TSPO with S2R might influence the binding of S2R to its ligands and also the interaction of S2R with PGRMC1 [91].

### 6.2. Regulatory Role of PGRMC1 in Heme Synthesis and Heme Trafficking

Heme is an essential cofactor necessary for a plethora of critical biological functions, such as oxygen transport and delivery, the electron transport chain involving cytochromes, metabolism of steroids, lipids, and drugs involving cytochrome P450s, redox homeostasis, tryptophan metabolism, and the generation of second messengers, such as NO, CO, and cGMP. It is, therefore, not surprising that the synthesis of this cofactor is a tightly regulated process. The synthesis of heme from succinyl-CoA and glycine as the starting molecules is carried out by the sequential actions of eight enzymes, some located in the cytoplasm and others within the mitochondria. This necessitates the trafficking of intermediary metabolites between the two cellular compartments. The first enzyme, δ-aminolevulinate synthetase (two isoforms: ALAS1 and ALAS2), is the rate-limiting enzyme that is negatively regulated by heme, the final end product of the synthetic pathway. The terminal enzyme, ferrochelatase, incorporates iron into protoporphyrin IX to generate heme. Both of these enzymes are located in the mitochondria.

PGRMC1 is a heme-binding protein and interacts with several heme-containing proteins, such as cytochrome P450s. But, the demonstration of PGRMC1 as a modulator of ferrochelatase [92] places PGRMC1 right at the center of iron/heme biology. Piel et al. [92] were able to demonstrate that PGRMC1 and ferrochelatase interact with each other and that this interaction facilitates the release of heme from ferrochelatase for the subsequent binding to PGRMC1. As such, it appears that PGRMC1 might serve as an intermediary in delivering heme from the terminal enzyme in the heme-synthetic pathway to other heme-binding proteins, such as cytochromes. Accordingly, the effect of PGRMC1 on the catalytic activity of ferrochelatase depends on whether or not PGRMC1 is present in the heme-free form or heme-bound form. When not bound to heme, PGRMC1 forms a complex not only with ferrochelatase but also with transferrin receptor–transferrin in endosomes, thus possibly facilitating the delivery of iron from transferrin to the reaction mediated by ferrochelatase to form heme. Furthermore, the release of heme from ferrochelatase represents a rate-limiting step in the catalytic activity of the enzyme; this release cannot occur in the absence of heme-free PGRMC1 to accept heme from ferrochelatase. In this manner, heme-free PGRMC1 stimulates the enzymatic activity of ferrochelatase. When heme is generated, it binds to PGRMC1. Now, the heme-bound PGRMC1 functions as a heme sensor and dissociates from ferrochelatase, thus suppressing the iron delivery and the enzymatic activity of ferrochelatase. Inhibitors of PGRMC1, such as AG-205, interfere with this regulatory process and thereby block not only the synthesis of heme but also the trafficking of heme from the mitochondria to heme-binding proteins at other cellular locations.

### 6.3. Role of PGRMC1–Heme Complex in the Function of EGFR and Its Relevance to Cancer and Chemosensitivity

The role of epidermal growth factor (EGF) signaling via its receptor in the plasma membrane (EGFR) in cancer cell proliferation is well known [93]. The EGFR signaling pathway is overactive in several cancers (e.g., lung cancer, colon cancer) and a number of selective inhibitors of EGFR tyrosine kinase are in clinical use for cancer therapy [94]. Therefore, it is important to understand the factors that are responsible for the overactivation of EGFR in cancer. PGRMC1 represents one such factor [62]. PGRMC1 dimerizes in the presence of heme as its ligand, and the resultant dimer of the heme-bound PGRMC1 interacts with EGFR and activates the downstream signaling pathways. The relevance of this regulatory process to cancer becomes significant not only because EGFR activation means the potentiation of tumor growth but also because cancer cells are known to overexpress PGRMC1 and accumulate iron/heme. This provides an ideal environment for the activation of EGFR by the PGRMC1–heme complex to promote cancer. Interestingly, the sixth coordination of heme needs to be free for the dimerization of PGRMC1–heme, as evident from the disruption of dimerization in the presence of CO, which binds to heme-Fe^2+^ to fulfill the sixth coordination. This corroborates the findings that CO is an inhibitor of tumor growth because CO will interfere with the formation of the PGRMC1–heme dimer that is necessary for EGFR activation.

PGRMC1 also functions as a heme chaperone. It promotes the trafficking of heme from the site of production (i.e., mitochondria) to heme-binding proteins in other locations. Cytochrome P450s represent the principal enzymatic machinery responsible for the inactivation of a variety of chemotherapeutic agents. It has been demonstrated that PGRMC1 traffics heme to some of these cytochrome P450s [62,75,76]. In the same study that demonstrated the involvement of PGRMC1 in EGFR activation, the authors provided evidence of the delivery of heme to two specific cytochrome P450s, namely CYP2D6 and CYP3A4 [62]. In other words, PGRMC1 plays an important role as a critical determinant of the catalytic activity of these cytochrome P450s. The relevance of this finding to cancer becomes obvious given that these two cytochrome P450s are involved in the inactivation of the chemotherapeutic agent doxorubicin. As such, the overexpression of PGRMC1 in cancer cells enhances metabolism and the consequent inactivation of doxorubicin, thereby contributing to chemoresistance. With a similar logic, the same can be said about excess iron/heme in cancer cells as a potential cause of the development of resistance to selective chemotherapeutic agents.

### 6.4. Function of PGRMC1 in Hepcidin Expression and Its Relevance to Hemochromatosis

Circulating levels, as well as tissue levels, of iron are regulated by two hormones, namely hepcidin and erythroferrone [95]. Several tissues and cell types are known to express these two hormones; however, the principal site of production is the liver for hepcidin and erythroblasts in the bone marrow for erythroferrone. These two hormones play opposite roles in the maintenance of iron levels in plasma and tissues. Hepcidin decreases and erythroferrone increases the circulating levels of iron. The biological activity of erythroferrone is mediated by changes in hepcidin expression. As expected from the opposing effects of the two hormones, erythroferrone acts on the liver to suppress the production of hepcidin. The expression of hepcidin in hepatocytes is regulated via multiple signaling pathways [96,97]. The target for erythroferrone action is the BMP signaling pathway, which is suppressed by erythroferrone, thus resulting in a decreased production of hepcidin [98]. In this process, erythroferrone functions as a trap for BMPs by binding to them, thus preventing their interaction with membrane-bound BMP receptors. Even though hepcidin is often referred to as a hormone, it does not work in a manner similar to other conventional hormones that function either via cell–surface receptors with specific intracellular second messengers as the effectors or via nuclear receptors, which modulate transcription. In contrast, hepcidin interacts with ferroportin, the only transporter known in mammalian cells with the ability to export iron out of the cells [99,100]. The binding of hepcidin to ferroportin results in internalization and the subsequent degradation of ferroportin. Ferroportin is responsible for the release of dietary iron from enterocytes into plasma and also for the release of iron from macrophages following phagocytosis and the degradation of aged and damaged erythrocytes. These are the two major sources of iron in circulation. Hepcidin interferes with ferroportin function at these two sites, thus causing a decrease in the circulating levels of iron. Erythroferrone has the opposite effect because it suppresses hepcidin. Erythropoietin, the major hormone from the kidney that promotes erythropoiesis, induces the expression of erythroferrone from erythroblasts, which results in an increased availability of iron in plasma to support erythropoiesis.

PGRMC1 has been shown to be an inducer of hepcidin synthesis in hepatocytes [101]. The physiologically and clinically relevant agonists of PGRMC1 that elicit this effect are progesterone and mifepristone. An analysis of the signaling pathway responsible for this phenomenon has yielded surprising results. The effect of PGRMC1 is mediated by tyrosine kinases of the SRC family [101]. This is a surprising finding because until then only the SMAD1/5 pathway for BMP and the JAK/STAT3 pathway for IL-6 were known as the prominent regulators of hepcidin expression. There is no information available from published reports as to whether PGRMC1 has any influence on erythroferrone production.

Based on these findings, we can conclude that the activation of PGRMC1 increases the levels of hepcidin in plasma. This has indeed been demonstrated in mice with mifepristone administration and in women with progesterone administration [101]. Therefore, PGRMC1 is expected to decrease the plasma levels of iron resulting from the suppression of iron release from enterocytes and macrophages. Combined with the already established regulatory functions of PGRMC1 in heme synthesis, hemoglobinization in erythroid cells, and its biological feature as a heme sensor [92], the ability of this protein to regulate hepcidin production and iron levels in plasma suggest that PGRMC1 serves as a critical regulator of iron homeostasis and potentially even erythropoiesis.

Hemochromatosis is a genetic disorder arising from loss-of-function mutations in the iron-regulatory protein called HFE (High Fe or histocompatibility-like protein involved in Fe regulation). It is an iron-overload disease associated with increased levels of iron in circulation and a consequent increased uptake and accumulation of iron in most organs, including the liver, intestine, colon, and retina [52,53,88,102,103]. Interestingly, the iron levels in the duodenum and macrophages are lower than normal in this disorder. This points to the involvement of ferroportin in this phenomenon. Iron absorption occurs primarily in the duodenum and the absorbed dietary iron is released from the duodenal enterocytes into plasma by ferroportin. As mentioned earlier, the same transporter is also responsible for the release of iron from macrophages into plasma for the reutilization of hemoglobin-derived iron subsequent to the phagocytosis of aged/damaged erythrocytes and the degradation of heme. The mutant HFE disrupts the iron-sensing mechanism in hepatocytes, thus causing the suppression of hepcidin expression and release. The resultant decrease in hepcidin in plasma is responsible for the increased activity of ferroportin in duodenal enterocytes and macrophages. As a consequence, dietary iron absorption is increased and the macrophage release of iron is also increased in hemochromatosis, leading to an elevation of circulating iron. However, the same process causes a decrease in cellular levels of iron in these two cell types. Now, the question is could the activation of PGRMC1 in the liver be therapeutically relevant for the treatment of hemochromatosis? Currently, phlebotomy is commonly used to reduce iron levels in this disorder. But, what if hepcidin expression in the liver could be increased as a means to reverse the increased function of ferroportin in enterocytes and macrophages seen in hemochromatosis? If this can be achieved, it will decrease the absorption of dietary iron and also suppress the release of iron from macrophages, thus reversing the iron overload that is associated with hemochromatosis. It has been shown that the decrease in the hepatic expression of hepcidin in hemochromatosis is due to interference with BMP signaling [101]. Since PGRMC1 promotes the hepcidin expression in hepatocytes by a mechanism that does not involve BMP signaling, it is conceivable that endogenous and/or exogenous activators of PGRMC1 would be able to increase hepcidin production in the liver in patients with hemochromatosis as a novel pharmacological treatment strategy.

### 6.5. PGRMC1-Mediated Protection against Ferroptosis and Its Relevance to Cancer

Hepcidin is also produced in cancer cells [104,105]. This locally generated hepcidin suppresses the levels of ferroportin in the plasma membrane of these cells in an autocrine manner, thus leading to an increase in cellular levels of iron, one of the obligatory nutrients for supporting cell proliferation. Since the PGRMC1 expression has been shown to be increased in most cancers [106,107,108,109,110], it is possible that the hepcidin expression is induced in cancer cells by PGRMC1 as it has been shown in the liver. This could be at least one of the mechanisms underlying the tumor-promoting function of PGRMC1. When cancer cells are obligated to accumulate iron at higher levels to support their proliferation and growth, they also face the risk of ferroptosis at the same time. Cancer cells find several ways to prevent this cell death process as a means of survival, and one such mechanism involves PGRMC1 [106,107]. The overexpression of PGRMC1 in cancer cells decreases the labile iron pool, thus protecting the cells from iron-induced ferroptosis [111]. Though the exact mechanism underlying this phenomenon is not known, it is tempting to speculate that the regulation of heme synthesis by PGRMC1 could be involved in the process. If PGRMC1 facilitates the chelation of free iron into heme at the ferrochelatase-mediated step, it could decrease the labile iron pool. As such, PGRMC1 could not only help cancer cells accumulate iron via hepcidin but also protect them from iron-induced cell death. These findings highlight the possibility that the pharmacological blockade of PGRMC1 could be beneficial as a treatment strategy for cancer.

### 6.6. PGRMC2 as a Heme-Chaperone to Nucleus

Though PGRMC2 is related to PGRMC1 in amino acid sequence and binds heme as PGRMC1 does, relatively less is known about the biology of PGRMC2. The study that reported on the critical role of PGRMC1 in heme synthesis via its interaction with ferrochelatase also found evidence for the interaction of PGRMC2 with ferrochelatase [92]. However, the relevance of PGRMC2 in heme synthesis or heme chaperoning was not pursued further in that study. A recent study [112] has shown that PGRMC2 does function as a heme chaperone in a manner similar to PGRMC1 but the target proteins for heme delivery are different. PGRMC1 seems to largely target cytochrome P450s in the endoplasmic reticulum and mitochondria. In contrast, PGRMC2 targets the nuclear proteins Rev-Erbα and BACH1 for heme delivery. Interestingly, Rev-Erbα and BACH1 are heme-responsive transcriptional suppressors [113,114]. Heme is essential for the biological activity of these two proteins but, at the same time, the binding of heme promotes their proteasomal degradation as a negative-feedback regulation of the proteins as heme sensors. The function of PGRMC2 as a heme chaperone to nuclear proteins agrees with its localization in the nuclear membrane [112]. PGRMC2 is also present in the endoplasmic reticulum, suggesting that the protein might supply heme to at least some of the hemoproteins located in this membrane. Interestingly, the heme supplied by PGRMC2 to nuclear proteins comes from the mitochondria. Therefore, the earlier observations that ferrochelatase interacts with PGRMC2 [92] might be directly connected to this phenomenon. The findings that PGRMC2 functions as a heme chaperone to Rev-Erbα and BACH1 originated from research on brown adipocyte differentiation. In this regard, it is important to note that Rev-Erbα is a critical player in adipocyte differentiation. Heme is a promoter of adipogenesis and this effect is mediated by Rev-Erbα, a heme-binding transcriptional regulator [115]. BACH1 also might play an essential role in brown adipocyte differentiation. When PGRMC2 is deleted in adipocytes, mitochondrial heme levels are decreased, suggesting that the catalytic activity of ferrochelatase might be controlled in some manner by PGRMC2; this could happen if PGRMC2 functions as the acceptor of heme from ferrochelatase, thereby controlling the catalytic activity of the latter [92].

Heat production in brown adipocytes is a phenomenon that occurs in the mitochondria via the uncoupling protein UCP1. PGRMC2 has been shown to be critical for thermogenesis in brown adipocytes [112]. This might involve the regulation of the UCP1 expression by PGRMC2 because the deletion of PGRMC2 in adipocytes decreases UCP1. Thermogenesis in brown adipocytes is an integral part of energy homeostasis and defects in this process are associated with obesity and metabolic syndrome. It is of interest to note that non-alcoholic fatty liver, insulin resistance, diabetes, and other features of metabolic syndrome are a common occurrence in patients with the iron-overload disease hemochromatosis [116,117].

## 7. Conclusions

This present review highlights the biological functions of sigma receptors that are primarily relevant to iron/heme homeostasis and hence critical in diseases, such as hemochromatosis and cancer. The functions of these receptors that are relevant to their role in the regulation of iron/heme biology are summarized in Table 2. There has been significant confusion and uncertainty in the past regarding the molecular nature of S2R, but most of this has been clarified in recent years. S2R, PGRMC1, and PGRMC2 are distinct molecular entities that might elicit their biological effects independently or as distinct complexes among themselves. S1R stands alone in this aspect, independent of the other three proteins, in spite of several common features among them, such as progesterone binding and overlapping ligand selectivity. Nonetheless, all four proteins certainly play an important role in iron/heme biology, thus unraveling a novel aspect of iron/heme homeostasis. As such, these proteins represent actionable drug targets for the treatment of diseases, such as hemochromatosis and cancer.

## Figures and Tables

**Figure 1 ijms-24-14672-f001:**
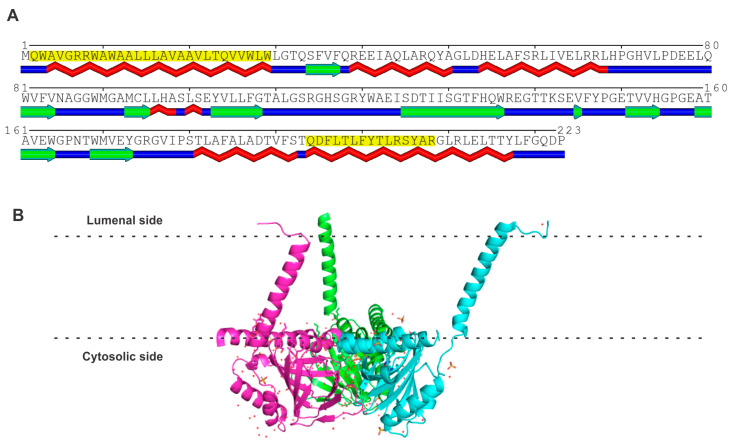
(**A**) Amino acid sequence and structure of human S1R. The predicted transmembrane domains (shaded in yellow), α-helices (indicated in red below the amino acid sequence), and β-strands (indicated in green below the amino acid sequence) according to the analysis of the amino acid sequence of human S1R [23]. (**B**) The homotrimeric structure of human S1R (PDB: 5HK1), each monomer with a membrane-spanning transmembrane domain at the N-terminus, and a second predicted transmembrane domain at the C-terminus on the membrane interface with the cytosolic side.

**Figure 2 ijms-24-14672-f002:**
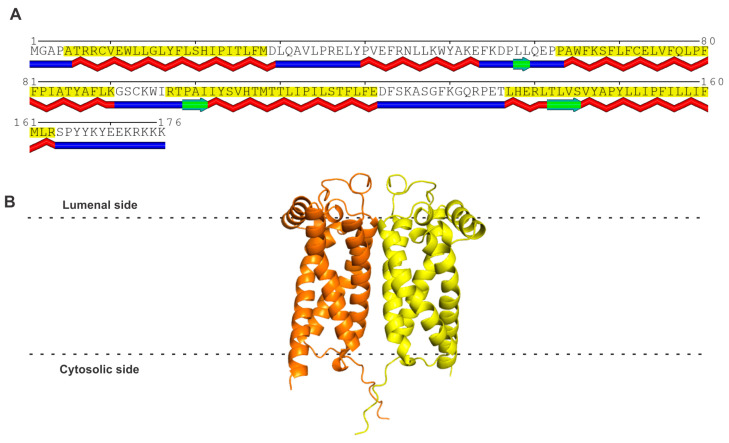
(**A**) Amino acid sequence and structure of human S2R. The predicted transmembrane domains (shaded in yellow) and β-strands (indicated in green below the amino acid sequence) according to the analysis of the amino acid sequence of human S2R using the POLYVIEW program [65]. (**B**) The AlphaFold model of human S2R is a monomer, but this structure was superimposed onto the recently described homodimeric structure of bovine S2R to generate the model for human S2R.

**Figure 3 ijms-24-14672-f003:**
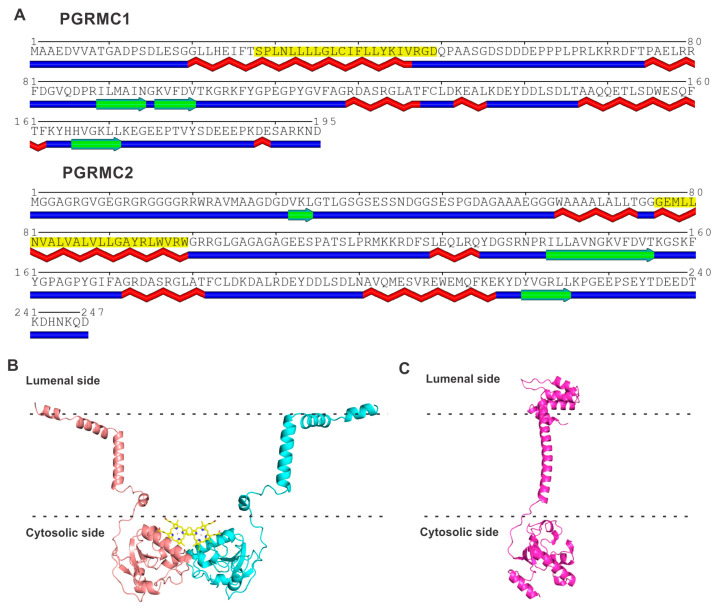
Amino acid sequence and structure for PGRMC1 and PGRMC2. (**A**) Transmembrane and secondary structure prediction of PGRMC1 and PGRMC2. The region highlighted in yellow in each protein represents the membrane-spanning transmembrane domain. Predicted α-helices are identified in red below the amino acid sequence, and β-strands are identified in green below the amino acid sequence. The POLYVIEW program [65] was used for these predictions. (**B**) Robetta model for PGRMC1 homodimer based on the crystal structure (PDB: 4X8Y). The heme ligand bound to each monomer is shown in yellow. (**C**) Robetta model for PGRMC2 monomer. Membrane boundaries were predicted with OPM (Orientations of Proteins in Membranes) server [67].

**Table 1 ijms-24-14672-t001:** Amino acid sequence identity among S1R, S2R, PGRMC1, and PGRMC2 determined using the multiple sequence alignment program Clustal Omega.

	S1R (%)	S2R (%)	PGRMC1 (%)	PGRMC2 (%)
**S1R**	100	21	24	25
**S2R**	21	100	21	21
**PGRMC1**	24	21	100	58
**PGRMC2**	25	21	58	100

**Table 2 ijms-24-14672-t002:** Functions of S1R, S2R, PGRMC1, and PGRMC2 that are relevant to their role in the regulation of iron/heme homeostasis.

**S1R**
Heme-binding protein (theoretical prediction); Activates NRF2 signaling and increases cellular levels of glutathione and NADPH; Decreases labile iron pool; Decreases lipid peroxidation and protects against ferroptosis.
**S2R**
Heme-binding protein (theoretical prediction); Negatively regulated by p53, a hemoprotein; Increases cellular levels of cholesterol, which positively correlates with heme levels.
**PGRMC1**
Heme-binding protein; Heme-dependent dimerization; Regulation of EGFR and cytochrome P450s by heme-bound homodimer; Regulation of heme synthesis via interaction with ferrochelatase; Chaperoning heme to specific hemoproteins (e.g., cytochrome P450s); Induction of hepcidin production; Protection against ferroptosis; CO sensor; Interaction with S2R and LDLR to increase cellular uptake of LDL-cholesterol.
**PGRMC2**
Heme-binding protein; Potential regulation of heme synthesis via interaction with ferrochelatase; Chaperoning heme to nuclear heme-responsive transcription factors; Chaperoning heme to specific cytochrome P450s; Uncoupling of electron transport chain from oxidative phosphorylation via induction of UCP1; Increase in thermogenesis.

## Data Availability

This is a review article. All the data contained in this article are freely available to the scientific community and the public.

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
