# Peer review of "Sigma Receptors: Novel Regulators of Iron/Heme Homeostasis and Ferroptosis"

_ijms, 2023, doi:10.3390/ijms241914672_

Round 1

Reviewer 1 Report

Review comments on ijms-2553093

Journal: International Journal of Molecular Sciences

Manuscript ID: ijms-2553093

Type of manuscript: Review

Title: Sigma Receptors: Novel Regulators of Iron/heme Homeostasis and Ferroptosis 

Authors: Nhi T. Nguyen, Valeria Jaramillo-Martinez, Marilyn Mathew, Varshini V. Suresh, Sathish Sivaprakasam, Yangzom D. Bhutia, Vadivel Ganapathy *

Major comments

(1)   In this review article, authors reviewed on four major members of sigma receptors, i.e., S1R, S2R, PGMC1, and PGMC2. It is now established that sigma receptors are non-opiate/non-phencyclidine receptors that bind progesterone and/or heme and several unrelated xenobiotics/chemicals. They reside in plasma membrane and in membranes of endoplasmic reticulum, mitochondria, and nucleus. Until recently, the biology/pharmacology of these proteins focused primarily on their role in neuronal functions in brain/retina. However, there have been recent developments in the field with the discovery of unexpected roles for these proteins in iron/heme homeostasis. 

(2)   Authors discussed on the structural features of these proteins based on the most recent discoveries in the field and then to focus on the interesting roles of these proteins in iron/heme metabolism and homeostasis. The regulatory functions of sigma receptors in the biology of iron and heme are among the latest findings in the field and have not been the major focus of any of previous review articles that have been published to date on these proteins.

(3)   This review article contains enough numbers of previous studies and well documented focusing on iron/heme homeostasis. This review article is very interesting and contains many pieces of information worth for the researchers working in this field. Further, even for the readers in other fields, this review might be very interesting and, therefore, this manuscript seems potentially acceptable for publication in IJMS.

(4)   However, before the final decision to be made, I would like to clarify following points.

(5)   Authors focused on PGRMC1 and PGRMC2, particularly on their regulatory role of iron/heme homeostasis based on the results reported by ref. #62 (Kabe et al., (2016)) (page 8, lines 334-341; page 9, lines 380-389; pages 11-12, lines 506-517). In this scenario, CO seems important role to make dissociation of the heme stacked dimer into a monomer. It must be noted, however, that the binding of CO to the heme moiety requires its ferrous state. However, there is no such mechanism being mentioned to maintain the PGRMC1 heme moiety in reduced state in this review or other previous studies.  

(6)   It is also not clear why authors did not mention about their following studies (Kabe et al. (2021), Cancers, 13, 3265) or their short review (Kabe et al., (2018), J. Clin. Biochem. Nutr., 63, 12-17)?

(7)   On the other hand, PGRMC1 is known to interact with ferrochelatase strongly and may serve as an intermediary in delivering heme from the terminal enzyme in the heme-synthetic pathway (i.e., ferrochelatase) to other heme-binding proteins such cytochromes (ref. #86). This scenario is more likely to occur and PGRMC2 would have a similar native role. 

(8)   In this context, I also do not understand why authors did not mention an important study conducted by McGuire et al. (2021), (J. Biol. Chem., 297, 101316)? Other Espenshade group’s works are also not cited.

(9)   It seems this review does not argue deeply on PGRMC1 and PGRMC2 in their roles of steroid binding ability.

Minor comments

(1)   “Amino acid sequence identity” might be better than “Sequence identity” (page 2, line 88).

(2)   Reference number is not indicated (page 10, line 456).

(3)   A period should be a comma. (page 13, line 580)

(4)   Authors’ names are missing for ref #84 (page 18, line 859).

Author Response

We thank the reviewer for his/her valuable comments.

We have now added a new paragraph describing potential mechanisms that might play a role in the transition between heme-ferric and heme-ferrous in heme-dimerized homodimer of PGRMC1. This strengthens the postulated function of PGRMC1 as a CO sensor.

We have now added additional references pertaining to publications by Kabe et al. in this area.

Yes, we agree in that PGRMC2 might have a role in the regulation of ferrochelatase similar to that of PGRMC1. This has been indicated in the original version while discussing PGRMC2.

We have now added new references by McGuire and Espenshade, both describing the potential role of PGRMC1 as a CO sensor.

As indicated in the Introduction, the current review primarily focused on the role of sigma receptors in iron/heme homeostasis. Therefore, we did not elaborate the role of these proteins as steroid-binding proteins. There are a number of excellent reviews covering this particular topic.

The minor comments have been taken care of.

Reviewer 2 Report

This is an interesting work that reviews the unexpected roles of the rather mysterious sigma receptors in the iron/heme metabolism. It describes the structural properties of the 4 receptors, their heme binding and direct and indirect effects on heme regulation and on hepcidin expression, that were shown to have effects on ferroptosis in some cancer models. The review is well-organized and written.

I think it is good and can be accepted after the two minor modifications I suggested. It reviews the properties of the sigma receptors to show that they have unexpected roles in the regulation of heme/iron regulation. This is novel and relevant for people working in the field.
The references semm appropriate
The tables and figures are only about the structures of the 4 proteins, additional data on their functions would improve the review

 -   I suggest including a table summarizing the main characteristics of the receptors.

 -   Line 617. “Overexpression of PGRMC1 in cancer cells increases the labile iron pool, thus protecting the cells from iron-induced ferroptosis [105].” PGRMC1 should decrease the labile iron pool. Verify. 

Author Response

We thank the reviewer for his/her valuable comments.

We have now added a new Table (Table 2) summarizing the functions of the sigma receptors that are relevant to their role in the regulation of iron/heme homeostasis.

We corrected the error. Thank you for pointing out this mishap.

Round 2

Reviewer 1 Report

Review comments on ijms-2553093-revised version

Journal: International Journal of Molecular Sciences

Manuscript ID: ijms-2553093-revised version

Type of manuscript: Review

Title: Sigma Receptors: Novel Regulators of Iron/heme Homeostasis and Ferroptosis 

Authors: Nhi T. Nguyen, Valeria Jaramillo-Martinez, Marilyn Mathew, Varshini V. Suresh, Sathish Sivaprakasam, Yangzom D. Bhutia, Vadivel Ganapathy *

Major comments

(1)   In this review manuscript, authors reviewed on four major members of sigma receptors (S1R, S2R, PGMC1, and PGMC2). It is now established that sigma receptors are non-opiate/non-phencyclidine receptors that bind progesterone and/or heme and several unrelated xenobiotics/chemicals. Authors discussed on the structural features of these proteins based on the most recent discoveries by focusing on the interesting roles of these proteins in iron/heme metabolism and homeostasis. The regulatory functions of sigma receptors in the biology of iron and heme are among the latest findings in the field and have not been the major focus of any of previous review articles that have been published to date on these proteins.

(2)   In this revised version of the manuscript (ijms-2553093-revised version), authors made appropriate revisions against the comments and suggestions raised by the reviewer, as follows (3) ~(7).

(3)   Original reviewer comment : Authors focused on PGRMC1 and PGRMC2, particularly on their regulatory role of iron/heme homeostasis based on the results reported by ref. #62 (Kabe et al., (2016)) (page 8, lines 334-341; page 9, lines 380-389; pages 11-12, lines 506-517). In this scenario, CO seems important role to make dissociation of the heme stacked dimer into a monomer. It must be noted, however, that the binding of CO to the heme moiety requires its ferrous state. However, there is no such mechanism being mentioned to maintain the PGRMC1 heme moiety in reduced state in this review or other previous studies. 

Author response : We have now added a new paragraph describing potential mechanisms that might play a role in the transition between heme-ferric and heme-ferrous in heme-dimerized homodimer of PGRMC1. This strengthens the postulated function of PGRMC1 as a CO sensor.

Reviewer comment : It is an appropriate amendment and this revision will strengthens the postulated function of PGRMC1 as a CO sensor.

(4)   Original reviewer comment : It is also not clear why authors did not mention about their following studies (Kabe et al. (2021), Cancers, 13, 3265) or their short review (Kabe et al., (2018), J. Clin. Biochem. Nutr., 63, 12-17)?

Authors response : We have now added additional references pertaining to publications by Kabe et al. in this area.

Reviewer comment : I confirmed that authors added new references #72, #79, and #80 in the revised version.

(5)   Original reviewer comment : On the other hand, PGRMC1 is known to interact with ferrochelatase strongly and may serve as an intermediary in delivering heme from the terminal enzyme in the heme-synthetic pathway (i.e., ferrochelatase) to other heme-binding proteins such cytochromes (ref. #86). This scenario is more likely to occur and PGRMC2 would have a similar native role. 

Authors response : Yes, we agree in that PGRMC2 might have a role in the regulation of ferrochelatase similar to that of PGRMC1. This has been indicated in the original version while discussing PGRMC2.

Reviewer comment : I confirmed that 

(6)   Original reviewer comment : In this context, I also do not understand why authors did not mention an important study conducted by McGuire et al. (2021), (J. Biol. Chem., 297, 101316)? Other Espenshade group’s works are also not cited.

Authors response : We have now added new references by McGuire and Espenshade, both describing the potential role of PGRMC1 as a CO sensor.

Reviewer comment : I confirmed that new references by McGuire and Espenshade were added in the revised manuscript as #75 and #76.

(7)   Original reviewer comment : It seems this review does not argue deeply on PGRMC1 and PGRMC2 in their roles of steroid binding ability.

Authors response : As indicated in the Introduction, the current review primarily focused on the role of sigma receptors in iron/heme homeostasis. Therefore, we did not elaborate the role of these proteins as steroid-binding proteins. There are a number of excellent reviews covering this particular topic.

Reviewer comment : I understand the situation. This would be a clever way for review article.

(8)   This review article is very interesting by focusing on iron/heme homeostasis and contains many pieces of information worth for the researchers working in this field. Further, even for the readers in other fields, this review might be very interesting. Therefore, this manuscript seems now acceptable for publication in IJMS in the present form.

Reviewer 2 Report

The authors inserted the table as suggested, The manuscript improved